# RCFGL: Rapid Condition adaptive Fused Graphical Lasso and application to modeling brain region co-expression networks

**Souvik Seal**[1]*, **Qunhua Li**[2], **Elle Butler Basner**[2], **Laura M. Saba**[3], **Katerina Kechris**[1]

**1** Department of Biostatistics and Informatics, Colorado School of Public Health, University of Colorado Anschutz Medical Campus, Aurora, Colorado, United States of America, **2** Department of Statistics, Pennsylvania State University, University Park, Pennsylvania, United States of America, **3** Skaggs School of Pharmacy and Pharmaceutical Sciences, University of Colorado Anschutz Medical Campus, Aurora, Colorado, United States of America

* souvik.seal@cuanschutz.edu

**Data Availability Statement:** Associated software package can be found at this link, https://github.com/sealx017/RCFGL. All the codes and the extracted results from the simulation studies are

## Abstract

Inferring gene co-expression networks is a useful process for understanding gene regulation and pathway activity. The networks are usually undirected graphs where genes are represented as nodes and an edge represents a significant co-expression relationship. When expression data of multiple ($p$) genes in multiple ($K$) conditions (e.g., treatments, tissues, strains) are available, joint estimation of networks harnessing shared information across them can significantly increase the power of analysis. In addition, examining condition-specific patterns of co-expression can provide insights into the underlying cellular processes activated in a particular condition. Condition adaptive fused graphical lasso (CFGL) is an existing method that incorporates condition specificity in a fused graphical lasso (FGL) model for estimating multiple co-expression networks. However, with computational complexity of $O(p^2 K \log K)$, the current implementation of CFGL is prohibitively slow even for a moderate number of genes and can only be used for a maximum of three conditions. In this paper, we propose a faster alternative of CFGL named rapid condition adaptive fused graphical lasso (RCFGL). In RCFGL, we incorporate the condition specificity into another popular model for joint network estimation, known as fused multiple graphical lasso (FMGL). We use a more efficient algorithm in the iterative steps compared to CFGL, enabling faster computation with complexity of $O(p^2 K)$ and making it easily generalizable for more than three conditions. We also present a novel screening rule to determine if the full network estimation problem can be broken down into estimation of smaller disjoint sub-networks, thereby reducing the complexity further. We demonstrate the computational advantage and superior performance of our method compared to two non-condition adaptive methods, FGL and FMGL, and one condition adaptive method, CFGL in both simulation study and real data analysis. We used RCFGL to jointly estimate the gene co-expression networks in different brain regions (conditions) using a cohort of heterogeneous stock rats. We also provide an accommodating *C* and *Python* based package that implements RCFGL.

provided with detailed documentation. The real data can be accessed through GSE173141, https://www.ncbi.nlm.nih.gov/geo/query/acc.cgi?acc=GSE173141.

**Funding:** Q.L. was supported by the National Institute of General Medical Sciences (NIGMS) of the National Institute of Health (NIH) grant R01GM109453. E.B.B. was supported by the NIGMS training grant T32 GM102057 awarded to Pennsylvania State University. L.M.S. and K.K. were supported by the National Institute on Drug Abuse (NIDA) of the NIH under award numbers P30DA044223. L.M.S. was also supported by NIDA under award number P50DA037844 and by the National Institute on Alcohol Abuse and Alcoholism (NIAAA) of the NIH under award number R24AA013162. K.K. was also supported by the National Heart, Lung, and Blood Institute (NHLBI) of the NIH under award number R01HL152735. The funders had no role in study design, data collection and analysis, decision to publish, or preparation of the manuscript.

**Competing interests:** No competing interests declared.

## Author summary

Inferring gene co-expression networks can be useful for understanding pathway activity and gene regulation. While jointly estimating co-expression networks of multiple conditions, taking into account condition specificity, such as information about an edge being present only in a specific condition or an edge being present across all the conditions, substantially increases the power. In this paper, a computationally rapid condition adaptive method for jointly estimating gene co-expression networks of multiple conditions is proposed. The novelty of the method is demonstrated through a broad range of simulation studies and a real data analysis with multiple brain regions from a genetically diverse cohort of rats.

## Introduction

A gene co-expression network is an undirected graph, where each node corresponds to a gene, and gene pairs are connected with an edge if they share a significant co-expression relationship [1–3]. Gene co-expression network analysis is a useful tool for uncovering the complex molecular interplay in biological processes [4–7]. Fitting Gaussian graphical models (GGM) is a popular approach for constructing biological networks in various applications [8–13]. In the context of gene co-expression network analysis, GGM assumes a multivariate normal distribution between the expression profiles of a set of genes [14]. The estimate of the inverse of the covariance matrix (also known as "precision matrix") is then examined to find which pairs of genes have significant conditional dependence and the co-expression network is constructed based on the dependence structure. The nonzero off-diagonal elements of the estimated precision matrix represent edges in the network.

Numerous approaches [15–21] have focused on the estimation of the aforementioned precision matrix. In most realistic scenarios, the number of genes ($p$) is much larger than the number of samples ($n$). It compels the researchers to use some form of regularization to induce sparsity in the estimation of the $p$-dimensional precision matrix. Yuan and Lin [16], Banerjee et al. [17], Friedman et al. [18], considered a penalized maximum likelihood model with $\ell_1$ regularization, known as graphical lasso (GL). Solving the GL model is a constrained convex optimization problem. Alternating direction method of multipliers (ADMM) [22–28] is a widely popular algorithm for solving constrained convex optimization problems. Different variations of ADMM have been used to solve the GL problem [29–34].

In a multi-condition gene co-expression study, the co-expression profiles across multiple ($K$) conditions are available and it is of great interest to find out how similar or dissimilar the co-expression networks are across those conditions [35–37]. For example, a particular co-expression network module can be present in a tumor tissue but not in a healthy tissue and thus, can serve as a key tool in identification of the tissue-type. There are methods like DiffCoEx [38], DICER [39] and DINGO [40] which particularly aim to study such differential co-expression patterns between two conditions. Broadly, these methods compare the sample correlation of every pair of genes between two conditions. The problem with such approaches is two-fold: firstly, the sample correlation may not be an appropriate measure of co-expression in many datasets, especially with a large number of genes and a limited sample-size and secondly, with more than two conditions, the approaches compare the pairs of conditions independently and thereby failing to perform a joint comparison in the true sense. Alternatively, a joint analysis of co-expression networks (more generally, any graphical networks) harnessing shared information across different conditions can be significantly

more powerful than individual analyses [41, 42]. Fused graphical lasso (FGL) [43] is one of the most popular approaches for joint estimation of multiple graphical networks. As the name suggests, FGL is an extension of the GL model in the context of multiple conditions. It simultaneously estimates multiple precision matrices, corresponding to multiple conditions, by considering the sum of multiple GL likelihoods and further employs a standard lasso penalty [44] and a pairwise fused lasso penalty [45] across the conditions. The standard lasso penalty encourages sparsity in the network estimation and the pairwise fused lasso penalty ensures that the networks share some degree of similarity. A similar method named fused multiple graphical lasso (FMGL) was proposed by Yang et al. [46]. FMGL considers a sequential fused lasso penalty across the conditions instead of the pairwise penalty considered in FGL. FGL and FMGL are equivalent when there are only two conditions. Both the methods use the iterative ADMM algorithm [22] for estimating the parameters. However, FMGL makes use of a very efficient intermediate step originally described in Condat (2013) [47] that substantially speeds up the computation (from $O(K \log K)$ to $O(K)$).

The fused lasso penalties both pairwise and sequential inherently assume that the precision matrices and consequently, the co-expression networks in all conditions are equally similar to each other. This assumption is rigid and may easily be violated in most real data scenarios. For example, tissues of two different tumor sub-types are expected to be more similar to each other than a healthy tissue. To account for such condition specific similarities and dissimilarities into the FGL framework, Lyu et al. [48] developed condition adaptive fused graphical lasso (CFGL). The penalty term considered in CFGL is a modification of the pairwise fused lasso penalty that incorporates binary weight matrices capturing condition-specificity. CFGL uses an iterative ADMM algorithm for estimating the parameters. However, the CFGL *R* package is limited because it can only accommodate a maximum of three conditions and is prohibitively slow even for a moderate number of genes ($p \approx 1000$). Thus, in a dataset with more than three conditions and a large number of genes, the CFGL *R* package is not scalable.

In this paper, we propose a new method named rapid condition adaptive fused graphical lasso (RCFGL) for jointly estimating multiple co-expression networks that takes into account condition specificity, is computationally rapid, and can handle more than three conditions. Similar to CFGL, we compute the binary weight matrices that capture pairwise condition specificity. Instead of considering a pairwise fused lasso penalty, as considered in CFGL, we incorporate the computed weight matrices with a sequential fused lasso penalty. In that sense, RCFGL is a condition adaptive extension of the FMGL algorithm. We use iterative ADMM algorithm [22] for estimation of the parameters. As in FMGL, using a sequential fused lasso penalty enables us to solve an intermediate step efficiently using fast algorithms [47, 49, 50]. This particular step is one of the main reasons behind the computational hurdle faced in CFGL. The authors of FGL and FMGL both had proposed a set of necessary conditions that can be investigated prior to fitting the models to evaluate the existence of a block diagonal structure in the precision matrices to be estimated. It can drastically reduce the computational time since all the matrix operations of order $O(p^3)$ reduce to $\sum_{l=1}^{M} O(p_l^3)$ (where $M$ is the total number of blocks with *l*-th block having size $p_l$). We have theoretically shown that the same set of conditions can also be used in the context of RCFGL further facilitating the computation. Through extensive simulation studies, we verified the robustness of our proposed method and demonstrated the computational advantage. We also analyzed the gene co-expression networks of three different brain regions from a dataset of heterogeneous stock rats. Finally, we built a *C* and *Python* based package implementing RCFGL available here, https://github.com/sealx017/RCFGL.

## Materials and methods

Suppose there are $p$ genes with expression profiles available across $K$ conditions and there are $n_k$ individuals under each condition $k$. Let $y_i^k$ denote the $p \times 1$ expression vector corresponding to the $i$-th individual under condition $k$. $\mathbf{Y}^{(k)} = (y_1^k, \ldots, y_{n_k}^k)^T$ is the $n_k \times p$ matrix of expression vectors under condition $k$ and $\bar{\mathbf{Y}}^{(k)}$ is the corresponding $1 \times p$ column mean vector. It is assumed that $y_1^k, \ldots, y_{n_k}^k \in \mathbb{R}^p$, are independently and identically drawn from $N_p(\boldsymbol{\mu}_k, \boldsymbol{\Sigma}_k)$ where $\boldsymbol{\mu}_k \in \mathbb{R}^p$ and $\boldsymbol{\Sigma}_k \succ \mathbf{0}$ (the notation $\succ \mathbf{0}$ denotes positive-definiteness). Let $\boldsymbol{\Theta}^{(k)} = \boldsymbol{\Sigma}_k^{-1}$ denote the precision matrix under condition $k$. Upon estimating $\boldsymbol{\Theta}^{(k)}$, the gene co-expression network would be constructed by representing the genes as nodes and conditional dependencies as edges in a graph. To be more specific, two genes $i, j$ under condition $k$, will only be connected in the graph if and only if $\boldsymbol{\Theta}_{ij}^{(k)} \neq 0$. Throughout the paper we use $\mathbf{1}_n$ to denote $n \times 1$ vector of all $1$'s. Next, we discuss the existing methods for estimating $\boldsymbol{\Theta}^{(k)}$'s.

### Review of methods

**Fused graphical lasso and fused multiple graphical lasso.** Fused graphical lasso (FGL) [43] and fused multiple graphical lasso (FMGL) [46] maximize the following penalized log-likelihood function,

$$\underset{\boldsymbol{\Theta}^{(k)} \succ 0, k=1,\ldots,K}{\text{maximize}} \sum_{k=1}^{K} n_k [\log(\det(\boldsymbol{\Theta}^{(k)}) - \text{tr}(\mathbf{S}^{(k)}\boldsymbol{\Theta}^{(k)})] - \text{P}(\boldsymbol{\Theta}); \quad (1)$$

where $\mathbf{S}^{(k)} = (\mathbf{Y}^{(k)} - \mathbf{1}_{n_k}\bar{\mathbf{Y}}^{(k)})^T(\mathbf{Y}^{(k)} - \mathbf{1}_{n_k}\bar{\mathbf{Y}}^{(k)})/n_k$ is the sample covariance matrix and $\text{P}(\boldsymbol{\Theta})$ is the penalty term with $\boldsymbol{\Theta} = \{\boldsymbol{\Theta}^{(1)}, \ldots, \boldsymbol{\Theta}^{(K)}\}$. As mentioned earlier, the only difference between FGL and FMGL is in the penalty term, $\text{P}(\boldsymbol{\Theta})$. FGL considers a pairwise fused lasso penalty and FMGL considers a sequential fused lasso penalty which have the following forms,

$$\text{P}^{\text{FGL}}(\boldsymbol{\Theta}) = \lambda_1 \sum_{i \neq j} \sum_{k=1}^{K} |\boldsymbol{\Theta}_{ij}^{(k)}| + \lambda_2 \sum_{i \neq j} \sum_{k < k'}^{K} |\boldsymbol{\Theta}_{ij}^{(k)} - \boldsymbol{\Theta}_{ij}^{(k')}|;$$

$$\text{P}^{\text{FMGL}}(\boldsymbol{\Theta}) = \lambda_1 \sum_{i \neq j} \sum_{k=1}^{K} |\boldsymbol{\Theta}_{ij}^{(k)}| + \lambda_2 \sum_{i \neq j} \sum_{k=1}^{K-1} |\boldsymbol{\Theta}_{ij}^{(k)} - \boldsymbol{\Theta}_{ij}^{(k+1)}|;$$

where $\lambda_1, \lambda_2$ are non-negative tuning parameters. The first term of both $\text{P}^{\text{FGL}}(\boldsymbol{\Theta})$ and $\text{P}^{\text{FMGL}}(\boldsymbol{\Theta})$ is the lasso penalty used in the GL model [18] that controls the overall sparsity. The second term of both the penalties controls the similarity of the off-diagonal elements of the precision matrices between conditions. Note that the second term of $\text{P}^{\text{FMGL}}(\boldsymbol{\Theta})$ is different from that of $\text{P}^{\text{FGL}}(\boldsymbol{\Theta})$ since it only focuses on differences between two consecutive conditions. If there are only two conditions i.e., $K = 2$, $\text{P}^{\text{FGL}}(\boldsymbol{\Theta}) = \text{P}^{\text{FMGL}}(\boldsymbol{\Theta})$. For $K = 3$, writing the penalties as functions of $\lambda_1, \lambda_2$, we show that $\text{P}^{\text{FGL}}(\boldsymbol{\Theta}, \lambda_1, \lambda_2) \leq \text{P}^{\text{FMGL}}(\boldsymbol{\Theta}, \lambda_1, 2\lambda_2)$. For $K > 3$, we are able to establish a crude connection: $\text{P}^{\text{FGL}}(\boldsymbol{\Theta}, \lambda_1, \lambda_2) \leq \text{P}^{\text{FMGL}}(\boldsymbol{\Theta}, \lambda_1, \lfloor \frac{K^2}{4} \rfloor \lambda_2)$ (S1 Text). $\text{P}^{\text{FGL}}(\boldsymbol{\Theta})$ encourages the same level of similarity between all the pairs of conditions and $\text{P}^{\text{FMGL}}(\boldsymbol{\Theta})$ encourages the same level of similarity between each consecutive pair of conditions. However, these assumptions may be violated in practical scenarios. For example, two different subtypes of tumor tissues can be more similar to each other than to a healthy tissue. Therefore, ideally the penalty term should be such that it penalizes the difference between the tumor subtypes more than it penalizes the difference between one of the tumor subtypes and the healthy

tissue. Lyu et. al. [48] addressed this issue by incorporating a special weight term into $\mathrm{P}^{\mathrm{FGL}}(\boldsymbol{\Theta})$ which is discussed in the next section.

**Condition adaptive fused graphical lasso.**   Lyu et. al. [48] introduced binary screening matrices: $\mathbf{W}^{(kk')} = [[\mathbf{w}_{ij}^{(kk')}]]$ for $k \neq k'$ defined as,

$$
\mathbf{w}_{ij}^{(kk')} = \begin{cases} 1 & \boldsymbol{\Theta}_{ij}^{(k)} \text{ and } \boldsymbol{\Theta}_{ij}^{(k')} \text{ are non-differential between conditions } k \text{ and } k' \\[2mm] 0 & \boldsymbol{\Theta}_{ij}^{(k)} \text{ and } \boldsymbol{\Theta}_{ij}^{(k')} \text{ are differential between conditions } k \text{ and } k'. \end{cases}
$$

The weight matrices are included in $\mathrm{P}^{\mathrm{FGL}}(\boldsymbol{\Theta})$ to define a penalty function that takes into account condition-specificity,

$$
\mathrm{P}^{\mathrm{CFGL}}(\boldsymbol{\Theta}) = \lambda_1 \sum_{i \neq j} \sum_{k=1}^{K} |\boldsymbol{\Theta}_{ij}^{(k)}| + \lambda_2 \sum_{i \neq j} \sum_{k < k'}^{K} \mathbf{w}_{ij}^{(kk')} |\boldsymbol{\Theta}_{ij}^{(k)} - \boldsymbol{\Theta}_{ij}^{(k')}|.
$$

The weight matrices in real data are unknown. Lyu et. al. [48] estimated them by performing a hypothesis test [51] for evaluating differences between the conditions. The test determines if the $ij$-th entry of the precision matrices: $\boldsymbol{\Theta}^{(k)}$ and $\boldsymbol{\Theta}^{(k')}$ is differential. If the test is rejected, $w_{ij}^{kk'}$ is set to 0, otherwise, it is set to 1. Going back to the example of two tumor subtypes and a healthy tissue, suppose the $ij$-th element is non-differential between the tumor subtypes (let's denote them as condition 1 and 2) but is differential between each of the tumor subtypes and the healthy tissue (let's denote it as condition 3). The weight terms in this case will be, $\mathbf{w}_{ij}^{12} = 1$, $\mathbf{w}_{ij}^{23} = 0$, and $\mathbf{w}_{ij}^{13} = 0$. As a consequence, $\mathrm{P}^{\mathrm{CFGL}}(\boldsymbol{\Theta})$ will penalize the difference between the tumor subtypes but not the difference between one of the tumor subtypes and the healthy tissue for the $ij$-th element.

## Proposed method

**Model.**   We propose to maximize the penalized log-likelihood from (1) with a new penalty term. We consider the binary weight matrices: $\mathbf{W}^{(kk')}$ discussed in the last section and include them to $\mathrm{P}^{\mathrm{FMGL}}(\boldsymbol{\Theta})$ instead of $\mathrm{P}^{\mathrm{FGL}}(\boldsymbol{\Theta})$ as in CFGL. Thus, the penalty term we propose has the following form,

$$
\mathrm{P}^{\mathrm{RCFGL}}(\boldsymbol{\Theta}) = \lambda_1 \sum_{i \neq j} \sum_{k=1}^{K} |\boldsymbol{\Theta}_{ij}^{(k)}| + \lambda_2 \sum_{i \neq j} \sum_{k=1}^{K-1} \mathbf{w}_{ij}^{(kk+1)} |\boldsymbol{\Theta}_{ij}^{(k)} - \boldsymbol{\Theta}_{ij}^{(k+1)}|.
$$

We name the method rapid condition adaptive fused graphical lasso (RCFGL) due to the computational speed it offers over CFGL. Note that when $K = 2$, RCFGL is equivalent to CFGL (since, $\mathrm{P}^{\mathrm{CFGL}}(\boldsymbol{\Theta}) = \mathrm{P}^{\mathrm{RCFGL}}(\boldsymbol{\Theta})$). Denote the set of all weight matrices as, $\mathbf{W} = \{\mathbf{W}^{kk'} : k < k'\}$. For $K > 2$, writing the penalties as functions of $\lambda_1, \lambda_2, \mathbf{W}$, we show that $\mathrm{P}^{\mathrm{CFGL}}(\boldsymbol{\Theta}, \lambda_1, \lambda_2, \mathbf{W}) \leq \mathrm{P}^{\mathrm{RCFGL}}(\boldsymbol{\Theta}, \lambda_1, \lambda_2, \mathbf{W}^*)$, where $\mathbf{W}^* = \{\mathbf{W}^{*(kk+1)} : k = 1, \ldots, K-1\}$ is a set of slightly modified weight matrices (S1 Text).

All the methods discussed so far consider penalty functions which are sums of two individual penalties: the first one being the standard lasso penalty controlling overall sparsity and the second one controlling similarity between conditions. The methods differ from each other only in terms of the second penalty term. In the second penalty term, FGL and CFGL consider all possible pairwise differences between the conditions, whereas FMGL and RCFGL consider only sequential differences. CFGL and RCFGL take into account condition specificity by incorporating weights, whereas FGL and FMGL are not condition adaptive (Table 1).

**Table 1. Penalty functions used in different methods.** The different methods consider penalty functions which are sum of two individual penalties and they differ only in the second term. The table categories the second penalty term for each of the methods by whether the method is condition adaptive and whether it uses the sequential difference.

| Sequential Difference〈br〉Condition Adaptive | No | Yes |
|---|---|---|
| No | FGL: $\sum_{i \neq j} \sum_{k < k'}^{K} \lvert \Theta_{ij}^{(k)} - \Theta_{ij}^{(k')} \rvert$ | FMGL (RFGL*): $\sum_{i \neq j} \sum_{k=1}^{K-1} \lvert \Theta_{ij}^{(k)} - \Theta_{ij}^{(k+1)} \rvert$ |
| Yes | CFGL: $\sum_{i \neq j} \sum_{k < k'}^{K} \mathbf{w}_{ij}^{kk'} \lvert \Theta_{ij}^{(k)} - \Theta_{ij}^{(k')} \rvert$ | RCFGL: $\sum_{i \neq j} \sum_{k=1}^{K-1} \mathbf{w}_{ij}^{(kk+1)} \lvert \Theta_{ij}^{(k)} - \Theta_{ij}^{(k+1)} \rvert$ |

*In the developed *Python* package, we provide an implementation of FMGL that we refer to as RFGL.

**ADMM algorithm.** We use an iterative ADMM algorithm [22] to maximize the penalized log-likelihood. Our algorithm is very similar to that used in FGL [43] and CFGL [48] with a few key modifications. The algorithm requires several intermediate variables such as $\mathbf{Z}$, $\mathbf{U}$ that do not have any direct interpretation. We rewrite the problem as,

$$\underset{\Theta, \mathbf{Z}}{\text{minimize}} - \sum_{k=1}^{K} n_k [\log(\det(\Theta^{(k)}) - \text{tr}(\mathbf{S}^{(k)} \Theta^{(k)})] + \text{P}^{\text{RCFGL}}(\mathbf{Z});$$

incorporating the constraint of positive-definiteness: $\Theta^{(k)} \succ 0$ for $k = 1, \ldots, K$ and the constraint that $\mathbf{Z}^{(k)} = \Theta^{(k)}$ for $k = 1, \ldots, K$, where $\mathbf{Z} = \{\mathbf{Z}^{(1)}, \ldots, \mathbf{Z}^{(K)}\}$. The corresponding scaled augmented Lagrangian [22] can be written as,

$$L_\rho(\Theta, \mathbf{Z}, \mathbf{U}) \quad = - \sum_{k=1}^{K} n_k [\log(\det(\Theta^{(k)}) - \text{tr}(\mathbf{S}^{(k)} \Theta^{(k)})] + \text{P}^{\text{RCFGL}}(\mathbf{Z}) +$$
$$\frac{\rho}{2} \sum_{k=1}^{K} \lVert \Theta^{(k)} - \mathbf{Z}^{(k)} + \mathbf{U}^{(k)} \rVert_F^2 - \frac{\rho}{2} \sum_{k=1}^{K} \lVert \mathbf{U}^{(k)} \rVert_F^2$$

(2)

where $\mathbf{U} = \{\mathbf{U}^{(1)}, \ldots, \mathbf{U}^{(K)}\}$ are dual variables, $\rho$ is a penalty parameter and $\lVert . \rVert_F$ denotes the Frobenius norm.

The algorithm is as follows,

1. Initialize the variables: $\Theta^{(k)} = \mathbf{I}$, $\mathbf{Z}^{(k)} = 0$, $\mathbf{U}^{(k)} = \mathbf{0}$ for $k = 1, \ldots, K$.

2. Select a constant $\rho > 0$.

3. For $i = 1, 2, 3, \ldots$ until convergence:

   i. For $k = 1, \ldots, K$, update $\Theta_{(i)}^{(k)}$ as the minimizer (with respect to [w.r.t] $\Theta^{(k)}$) of
   $-n_k [\log(\det(\Theta^{(k)}) - \text{tr}(\mathbf{S}^{(k)} \Theta^{(k)})] + \frac{\rho}{2} \lVert \Theta^{(k)} - \mathbf{Z}_{(i-1)}^{(k)} + \mathbf{U}_{(i-1)}^{(k)} \rVert_F^2.$
   Let $\mathbf{V}\mathbf{D}\mathbf{V}^T$ denote the eigen-decomposition of $\mathbf{S}^{(k)} - \rho/n_k (\mathbf{Z}_{(i-1)}^{(k)} + \mathbf{U}_{(i-1)}^{(k)})$.
   The solution of the above minimization [52] is given by $\mathbf{V}\tilde{\mathbf{D}}\mathbf{V}^T$, where $\tilde{\mathbf{D}}_{jj}$ is the diagonal matrix with $j$-th diagonal element being

$$\rho/n_k \left( -\mathbf{D}_{jj} + \sqrt{\mathbf{D}_{jj}^2 + 4\rho/n_k} \right).$$

ii.  Update $\mathbf{Z}_{(i)}$ as the minimizer (w.r.t $\mathbf{Z}$) of

$$\mathrm{P}^{\mathrm{RCFGL}}(\mathbf{Z}) + \frac{\rho}{2}\sum_{k=1}^{K}||\boldsymbol{\Theta}_{(i)}^{(k)} - \mathbf{Z}^{(k)} + \mathbf{U}_{(i-1)}^{(k)}||_{F}^{2}$$

The problem can be rewritten as,

$$\underset{\mathbf{Z}}{\mathrm{minimize}}\left\{\mathrm{P}^{\mathrm{RCFGL}}(\mathbf{Z}) + \frac{\rho}{2}\sum_{k=1}^{K}||\mathbf{Z}^{(k)} - \mathbf{A}^{(k)}||_{F}^{2}\right\};\quad \mathbf{A}^{(k)} = \boldsymbol{\Theta}_{(i)}^{(k)} + \mathbf{U}_{(i-1)}^{(k)}$$

With the actual expression of $\mathrm{P}^{\mathrm{RCFGL}}(\mathbf{Z})$ the above problem takes the form,

$$\underset{\mathbf{Z}}{\mathrm{minimize}}\left\{\lambda_1\sum_{i\neq j}\sum_{k=1}^{K}|\mathbf{Z}_{ij}^{(k)}| + \lambda_2\sum_{i\neq j}\sum_{k=1}^{K-1}\mathbf{w}_{ij}^{(kk+1)}|\mathbf{Z}_{ij}^{(k)} - \mathbf{Z}_{ij}^{(k+1)}| + \frac{\rho}{2}\sum_{k=1}^{K}||\mathbf{Z}^{(k)} - \mathbf{A}^{(k)}||_{F}^{2}\right\}$$

The above problem is completely separable w.r.t each pair of matrix elements $(i, j)$, where $i \neq j$. It means that one can independently solve, for each pair $(i, j)$, the following minimization problem:

$$\underset{\mathbf{Z}_{ij}^{(1)},\dots,\mathbf{Z}_{ij}^{(K)}}{\mathrm{minimize}}\left\{\lambda_1\sum_{k=1}^{K}|\mathbf{Z}_{ij}^{(k)}| + \lambda_2\sum_{k=1}^{K-1}\mathbf{w}_{ij}^{(kk+1)}|\mathbf{Z}_{ij}^{(k)} - \mathbf{Z}_{ij}^{(k+1)}| + \frac{\rho}{2}\sum_{k=1}^{K}|\mathbf{Z}_{ij}^{(k)} - \mathbf{A}_{ij}^{(k)}|^{2}\right\} \quad (3)$$

The problem is known as the weighted 1-D fused lasso signal approximator, which can be solved very efficiently.

iii.  For $k = 1, \dots, K$, update $\mathbf{U}_{(i)}^{k}$ as $\mathbf{U}_{(i-1)}^{k} + (\boldsymbol{\Theta}_{(i)}^{k} - \mathbf{Z}_{(i)}^{k})$.

The step where using a sequential fused lasso penalty instead of a pairwise fused lasso penalty is beneficial is in Eq (3). When $\mathbf{w}_{ij}^{(kk+1)} = 1$ for all $k = 1, \dots, K - 1$, the problem of Eq 3 becomes the 1-D fused lasso signal approximator [45, 53] for which an efficient and exact solution is available by the algorithm of Condat et al. [47]. The *MATLAB* package of FMGL [46] also uses this particular algorithm. The algorithm of Condat et al. [47] treats the fused lasso signal approximator as a 1-D total variation denoising problem [54]. When $\mathbf{w}_{ij}^{(kk+1)} = 0$ for at least one $k$, the problem of Eq 3 can be thought of as a special case of a weighted 1-D total variation problem (where weights are 1 or 0). There is an efficient 'Taut-String' algorithm [49, 50] for solving weighted 1-D total variation denoising problems.

The algorithm of Condat et al. [47] and the 'Taut-String' algorithm [49, 50] both have computational complexity of $O(K)$ in most practical scenarios. Recall that FGL [43] uses a pairwise fused lasso penalty which results in the general fused lasso approximator [45, 53] in the $\mathbf{Z}$ updating step. FGL follows a path algorithm [53] for solving the above step which has computational complexity of $O(K \log K)$. In CFGL [48], the authors solve the $\mathbf{Z}$ updating step exactly for $K = 2$ and 3, but, do not provide any solution for $K > 3$. For details about the computation of the weight matrices: $\mathbf{W}^{kk+1}$ for $k = 1, \dots, K - 1$, we refer to the CFGL paper [48].

**Detecting block diagonal structure in the precision matrices.**  Here, we present a theorem involving a set of sufficient conditions that can be checked prior to fitting the ADMM algorithm and can potentially result in substantial computational benefit. A similar theorem has been used in the context of FGL [43] and FMGL [46]. Using the theorem, one would inspect the sample covariance matrices $\mathbf{S}^{(1)}, \dots, \mathbf{S}^{(K)}$ to determine if the solution to the RCFGL problem i.e., the estimates of the precision matrices: $\hat{\boldsymbol{\Theta}}^{(k)}$ for $k = 1, \dots, K$, are block-diagonal

after some permutation of the genes. The inspection is based on comparing the absolute values of $\mathbf{S}_{ij}^{(k)}$'s with the tuning parameter $\lambda_1$.

**Theorem 1** *Denote the set of p genes by C. Suppose, there are M many disjoint subsets of C s. t. $C_1 \sqcup C_2 \sqcup \ldots \sqcup C_M = C$. For the genes in $C_l$ to be completely disconnected from those in $C_{l'}$ in each of the resulting estimates, it will be sufficient to have $|n_k \mathbf{S}_{ij}^{(k)}| < \lambda_1$ for $k = 1, 2, \cdots, K, \forall i \in C_l, j \in C_{l'}$.*

Using Theorem 1, for a given value of $\lambda_1$, suppose we find out that the estimated precision matrices: $\hat{\mathbf{\Theta}}^{(k)}$ for $k = 1, \ldots, K$, will be block-diagonal with $M$ blocks i.e., they will have the following form,

$$\hat{\mathbf{\Theta}}^{(k)} = \begin{pmatrix} \hat{\mathbf{\Theta}}_1^{(k)} & \mathbf{0} & \mathbf{0} & \mathbf{0} \\ \mathbf{0} & \hat{\mathbf{\Theta}}_2^{(k)} & \mathbf{0} & \mathbf{0} \\ \mathbf{0} & \mathbf{0} & \ddots & \mathbf{0} \\ \mathbf{0} & \mathbf{0} & \mathbf{0} & \hat{\mathbf{\Theta}}_M^{(k)} \end{pmatrix} \tag{4}$$

where $\hat{\mathbf{\Theta}}_l^{(k)}$ for $k = 1, \ldots, K$ have the same dimensions and correspond to the same subset of genes: $C_l$. It would imply that instead of solving the RCFGL problem for full $\mathbf{\Theta}^{(k)}$, one can solve the RCFGL problems for $\mathbf{\Theta}_l^{(k)}$ for $l = 1, \ldots, M$ independently. This drastically reduces the computational complexity. Let the dimension of each block $\mathbf{\Theta}_l^{(k)}$ be $p_l \times p_l$ (the size of the subset $C_l$ is $p_l$), where $\sum_{l=1}^{M} p_l = p$. The ADMM algorithm discussed in the last section, involves eigen-decomposition of $K$ many $p \times p$ matrices which takes up computational complexity of $O(Kp^3)$. Whereas, solving block RCFGL problems will only have the computational complexity of $K \sum_{l=1}^{M} O(p_l^3)$. The proof of Theorem 1 can be found in S1 Text.

**Tuning parameter selection.** Following the suggestion of [43, 48] for selecting the tuning parameters $\lambda_1, \lambda_2$, we use an approximation of the Akaike information criterion (AIC),

$$AIC(\lambda_1, \lambda_2) = \sum_{k=1}^{K} [n_k \text{tr}(\mathbf{S}^{(k)} \hat{\mathbf{\Theta}}_{\lambda_1, \lambda_2}^{(k)}) - n_k \log(\det(\hat{\mathbf{\Theta}}_{\lambda_1, \lambda_2}^{(k)}) + 2E_k]$$

$\hat{\mathbf{\Theta}}_{\lambda_1, \lambda_2}^{(k)}$ is the precision matrix estimated for the $k$-th condition using the tuning parameters $\lambda_1$ and $\lambda_2$, and $E_k$ is the number of unique non-zero elements in $\hat{\mathbf{\Theta}}_{\lambda_1, \lambda_2}^{(k)}$. A grid search can then be performed to select $\lambda_1$ and $\lambda_2$ that minimize the $AIC(\lambda_1, \lambda_2)$ score. However, as pointed out by [43], such an approach may tend to choose models that are too large to be useful. Thus, in many cases, model selection is better guided by practical considerations, such as network interpretability and stability.

**Effect of ordering of the conditions.** Our penalty term, $\text{P}^{\text{RCFGL}}(\mathbf{\Theta})$ only considers sequential differences between the conditions. It implies that different ordering of the conditions would yield different penalty levels. For example, suppose there are three conditions: 1, 2 and 3, where the network of 1 is same as that of 3 but the network of 2 is totally different ($\mathbf{\Theta}^{(1)} = \mathbf{\Theta}^{(3)} \neq \mathbf{\Theta}^{(2)}$). If we consider the sequence $(1, 3, 2)$, $\text{P}^{\text{RCFGL}}(\mathbf{\Theta})$ will include the terms: $w_{ij}^{13}|\mathbf{\Theta}_{ij}^{(1)} - \mathbf{\Theta}_{ij}^{(3)}|$ that encourage similarity in the estimated networks of 1 and 3. But, if we consider the sequence $(1, 2, 3)$, $\text{P}^{\text{RCFGL}}(\mathbf{\Theta})$ will not include those terms, thereby not encouraging similarity between $\mathbf{\Theta}^{(1)}, \mathbf{\Theta}^{(3)}$. Thus, it can be potentially more powerful to use a particular ordering of conditions that places more similar conditions closer. We study the effect of misspecified ordering in our simulation studies.

The ordering can be based on biological information available about the degree of similarity across the conditions, for instance, relationship in cell lineages. Alternatively, we can use hierarchical clustering based on the gene expression data, $\mathbf{Y}^{(k)}$, $k = 1, \ldots, K$, and other sophisticated data-driven ways of obtaining a suitable ordering. As an example, here we discuss a simple method based on comparing the sample covariance matrices across the conditions. More specifically, we compute the sample covariance matrix for ever condition $k$, $\mathbf{S}^{(k)} = \left(\mathbf{Y}^{(k)} - 1_{n_k}\bar{\mathbf{Y}}^{(k)}\right)^T\left(\mathbf{Y}^{(k)} - \mathbf{1}_{n_k}\bar{\mathbf{Y}}^{(k)}\right)/n_k$. Then, we consider the Euclidean distance between a pair of conditions $(k, k')$ as,

$$d(k, k') = \sqrt{\sum_{i=1}^{n_k}\sum_{j=1}^{n_k}(\mathbf{S}_{i,j}^{(k)} - \mathbf{S}_{i,j}^{(k')})^2}.$$

We subject the generated distance matrix (between all the conditions) to hierarchical clustering to identify conditions that are closer or farther from each other and use these relationships to order the conditions in RCFGL. In our simulation studies, this procedure was able to detect the right ordering every time.

**Software implementation.** We mainly make use of Condat et. al.'s algorithm [47] available as a *c* code and a *Python* module named proxTV to build our package named RCFGL available here. We provide a *Jupyter* notebook [55] with detailed guidance for fitting the RCFGL model. Additionally, we provide an implementation of the FMGL model that we refer to as RFGL, an acronym for rapid fused graphical lasso. The order of the conditions can be specified by the users. We also provide functions for visualizing the estimated networks and compare them across conditions. The developed package can be found at this link, https://github.com/sealx017/RCFGL. All code used in the simulation studies of this manuscript are also provided with detailed documentation.

**Simulation setup.** We considered seven different simulation scenarios, (S1), (S2), ..., and (S7), with varying levels of differentiation across conditions described below. In each of the scenarios, we considered 500 genes and 100 subjects. Each gene co-expression network consisted of 5 equally sized sub-networks, each made of 100 genes.

1. In the first four simulation scenarios, our goal was to compare RCFGL with other three methods i.e., FGL, FMGL (referred to as, RFGL), and CFGL in terms of both estimation accuracy and computational time.

(a). In both (S1) and (S2), three conditions were considered i.e., $K = 3$. In (S1), the first two networks were exactly the same, whereas the third network shared only three sub-networks common with the first two and the other two sub-networks were generated independently. In (S2), the first two networks were again exactly the same but the third one did not share any sub-network common with the first two i.e., all 5 of its sub-networks were generated independently.

(b). In both (S3) and (S4), four conditions were considered i.e., $K = 4$. In (S3), the first two and the last two networks were the same as each other. In (S4), only the first two networks were the same and the other two were different.

2. In the last three simulation scenarios, we studied the effect of the ordering of the conditions on RCFGL's performance for three and four conditions. In (S5), three conditions were considered. The first and third conditions had the same networks, whereas the second network was entirely different. In (S6) and (S7), four conditions were considered. In (S6), the first and third networks were the same and the other two were different. In (S7), the first and fourth networks were the same and the other two were different.

$$(S1): K = 3, \Theta^{(1)} = \Theta^{(2)} = \begin{pmatrix} \Theta_1 & 0 & 0 & 0 & 0 \\ 0 & \Theta_2 & 0 & 0 & 0 \\ 0 & 0 & \Theta_3 & 0 & 0 \\ 0 & 0 & 0 & \Theta_4 & 0 \\ 0 & 0 & 0 & 0 & \Theta_5 \end{pmatrix}; \Theta^{(3)} = \begin{pmatrix} \Theta_1 & 0 & 0 & 0 & 0 \\ 0 & \Theta_2 & 0 & 0 & 0 \\ 0 & 0 & \Theta_3 & 0 & 0 \\ 0 & 0 & 0 & \Theta_4^{(3)} & 0 \\ 0 & 0 & 0 & 0 & \Theta_5^{(3)} \end{pmatrix}.$$

$(S2): K = 3, \Theta^{(1)} = \Theta^{(2)} \neq \Theta^{(3)}.$

$(S3): K = 4, \Theta^{(1)} = \Theta^{(2)} \neq \Theta^{(3)} = \Theta^{(4)}.$

$(S4): K = 4, \Theta^{(1)} = \Theta^{(2)} \neq \Theta^{(3)} \neq \Theta^{(4)}.$

$(S5): K = 3, \Theta^{(1)} = \Theta^{(3)} \neq \Theta^{(2)}.$

$(S6): K = 4, \Theta^{(1)} = \Theta^{(3)} \neq \Theta^{(2)} \neq \Theta^{(4)}.$

$(S7): K = 4, \Theta^{(1)} = \Theta^{(4)} \neq \Theta^{(2)} \neq \Theta^{(3)}.$

**Fig 1. The relationship between the precision matrices across conditions in different simulation scenarios.** Notice that in (S1), the first two conditions had the exactly same networks, whereas the network of the third condition was partially similar, sharing only the first three blocks. In (S2) and the following simulation scenarios, there was no such partial similarity considered and the conditions either shared the full network or they were entirely dissimilar.

All of the scenarios are summarized in terms of the precision matrices of different conditions in Fig 1. Next, we describe how the above networks and corresponding edge-weights were simulated.

To mimic real-world biological network structures [48, 56], we used the Barabasi-Albert model [57] to simulate the unweighted network topology, i.e., the adjacency matrix with indicator elements, 1 if an edge was present between a pair of genes and 0 otherwise. Next, the $k$-th weighted network, $A^{(k)}$ was generated as,

$$A_{ij}^{(k)} = \begin{cases} 1 & \text{if } i = j \\ 0 & \text{if } (i,j)\text{-th element of the } k\text{-th adjacency matrix was 0} \\ \sim \text{Unif}(D) & \text{if } (i,j)\text{-th element of the } k\text{-th adjacency matrix was 1} \end{cases}$$

where Unif($D$) refers to a uniform distribution with $D = [-0.9, -0.6] \cup [0.6, 0.9]$. To ensure that the weighted network was positive definite, an eigen-value adjustment was performed as, $A^{*(k)} = A^{(k)} + |\delta^{(k)}|\mathbf{I}$, where $\delta^{(k)}$ was the smallest eigen-value of $A^{(k)}$. Based on $A^{*(k)}$, covariance matrix $\Sigma_k = [\Sigma_{k(i,j)}]$ was constructed as,

$$\Sigma_{k(i,j)} = \frac{[A^{*(k)}]_{(i,j)}^{-1}}{\sqrt{[A^{*(k)}]_{(i,i)}^{-1}[A^{*(k)}]_{(j,j)}^{-1}}}.$$

With the covariance matrix $\Sigma_k$ (and, consequently $\Theta^{(k)}$), the gene expression vector of the $i$-th subject under condition $k$ was simulated as $y_i^{(k)} \sim N_p(0, \Sigma_k)$. Ten replications were considered in every simulation scenario and average findings were reported.

**Real data.** RNA expression from three brain regions was measured using high-throughput RNA sequencing on the Illumina HiSeq 4000 platform and a poly-A selection protocol (GSE173141). The original data set [58] included tissue from 88 alcohol and drug naïve heterogeneous stock rats and most rats had RNA-Seq libraries from all the regions. The average

number of raw reads per rat and brain region was 26.7 million. After extensive quality control, 83 rats remained with RNA-Seq from the lateral habenula (LHB) core, 84 rats with data from the infralimbic (IL) cortex, and 82 rats with data from the prelimbic (PL) cortex [59, 60]. Reads were trimmed to remove adaptors and low-quality base calls using cutAdapt [61]. They were then aligned to the Ensembl Rat transcriptome using RSEM (RNA-Seq Expectation-Maximization; [62]). An upper quantile scaling was initially applied to the estimated read counts for individual genes using the betweenLaneNormalization function from the EDASeq package in R [63]. A regularized (r)log was then used to transform the read counts using the DESeq2 package in *R* [64]. Finally, a batch effects adjustment was made using the ComBat function in the sva R package [65]. For this manuscript, we focused on 15,421 protein-coding genes common across all three brain regions and 64 rats having data available for all of those genes in all three brain regions. More details about the dataset can be found in S1 Text.

**Measures for evaluating performance.**   In the simulation studies, the estimation performance of the methods were assessed based on both network topology and edge-weights. Denote the true precision matrices as $\mathbf{\Theta}^{(k)}$ and the estimated precision matrices by $\hat{\mathbf{\Theta}}^{(k)}$ for $k = 1, \ldots, K$. We first determined the true and false positives and false negatives in the following way. If the $ij$-th edge was present in the true network of the $k$-th condition i.e., $\mathbf{\Theta}_{ij}^{(k)} \neq 0$ and was also identified in the estimated network i.e., $|\hat{\mathbf{\Theta}}_{i,j}^{(k)}| \geq tol$, it was counted as a true positive (TP). Here, *tol* was a chosen level of tolerance to define an edge and it was kept at 0.01. Similarly, if the edge was absent in the true network i.e., $\mathbf{\Theta}_{ij}^{(k)} = 0$ but was identified in the estimated network, it was counted as a false positive (FP). If the edge was present in the true network but was not identified in the estimated network i.e., $|\hat{\mathbf{\Theta}}_{i,j}^{(k)}| < tol$, it was counted as a false negative (FN). Next, the precision ($= \frac{\text{TP}}{\text{TP+FP}}$) and recall ($= \frac{\text{TP}}{\text{TP+FN}}$) were computed to plot the precision-recall curves. To judge the accuracy of edge-weight estimation, we computed the sum of squared error (SSE) between the estimated and the true precision matrices: $\sum_{k=1}^{K} \sum_{i=1}^{n_k} \sum_{j=1}^{n_k} (\mathbf{\Theta}_{i,j}^{(k)} - \hat{\mathbf{\Theta}}_{i,j}^{(k)})^2$. We compared the run-times of the methods based on a *MacOS* system with 32 GB RAM and *intel i9* CPU with 8 cores.

In the real data analysis, first we compared the run-times of the different methods. Next, we compared the estimation performance of RFGL and RCFGL to assess the advantages of condition adaptive estimation in this context. To demonstrate how similar the results from RCFGL and CFGL were, we inspected the top $Z$ edges of every brain region based on the absolute value of the estimated precision matrices. Let the sets of the top $Z$ edges detected by CFGL for regions LHB, IL and PL be respectively denoted by $M_1$, $M_2$ and $M_3$ and those by RCFGL be denoted as $N_1$, $N_2$ and $N_3$. In mathematical terms, we looked at the following proportion also known as the Jaccard index [66],

$$\text{prop}(Z) = \frac{|(\cup_{i=1}^{3} M_i) \cap (\cup_{i=1}^{3} N_i)|}{|(\cup_{i=1}^{3} M_i) \cup (\cup_{i=1}^{3} N_i)|} \tag{5}$$

for different values of $Z$. A value close to 1 for prop($Z$) would imply both the methods produced the same top $Z$ edges. As discussed earlier, the penalty terms of RCFGL and CFGL share a special inequality, $\text{P}^{\text{CFGL}}(\mathbf{\Theta}, \lambda_1, \lambda_2, \mathbf{W}) \leq \text{P}^{\text{RCFGL}}(\mathbf{\Theta}, \lambda_1, \lambda_2, \mathbf{W}^*)$, where $\mathbf{W}^*$ is a set of modified weight matrices defined as, $\mathbf{W}^* = (\mathbf{W}^{*12}, \mathbf{W}^{*23})$, where $\mathbf{W}^{*12} = \mathbf{W}^{12} + \mathbf{W}^{13}$ and $\mathbf{W}^{*23} = \mathbf{W}^{23} + \mathbf{W}^{13}$ (see Proposed method section and S1 Text). Thus, to achieve better agreement with CFGL, we fitted RCFGL with the modified set of weight matrices $\mathbf{W}^*$.

To examine the biological relevance of results from the network analysis by RCFGL, we first identified the hub-genes in every brain region, defined as the genes with more than five connections. Then, we checked which of those hub-genes had similar degree in the medial

prefrontal cortex regions IL and PL but different degree in the LHB region. Finally, we studied the functional enrichment separately for the two sets of genes: the ones whose degree decreased from IL and PL to LHB and the others whose degree increased from IL and PL to LHB. Functional enrichment was evaluated using the ShinyGO tool (version 0.76.2; http://bioinformatics.sdstate.edu/go/; [67]) specifying the KEGG Pathway and the Gene Ontology (GO) Biological Process database for simplicity. The pathways with at least three related genes and FDR < 0.05 were reported.

## Results

### Simulation study

We evaluated the performance of RCFGL in seven simulation scenarios described in Simulation setup section. For fitting FGL and CFGL, we used the corresponding *R* packages and for fitting FMGL (referred to as, RFGL) and RCFGL, we used our package. The methods RFGL, FGL, RCFGL, CFGL were denoted by different colored lines in all the figures. They were run with the same sets of hyperparameters, $(\lambda_1, \lambda_2)$. Different $\lambda_1$'s resulted in different numbers of edges detected, and different values of $\lambda_2$ modulated the similarity penalty from low to high.

**Simulation with three conditions.** The difference between (S1) and (S2) lied in the level of similarity across the networks. As discussed in Review of methods, FGL and RFGL both assume that networks of all the conditions share same level of similarity. Scenario (S1) was close to that assumption, whereas (S2) violated it since the third network did not share any similarity with the first two. Figs 2 and 3 respectively show the the precision-recall curves for

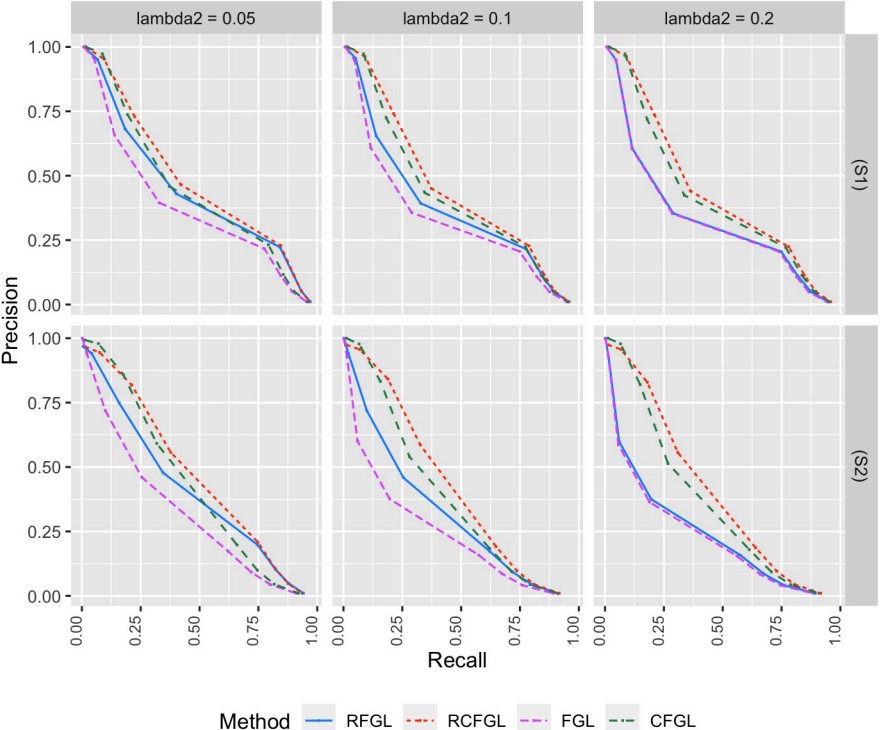

**Fig 2. Comparison of edge detection performance for simulations with three conditions.** Top and bottom rows respectively correspond to the precision-recall curves in scenario (S1) and scenario (S2). The *x* and *y* axes respectively correspond to recall and precision of the methods for different values of $\lambda_1$. Three different values of $\lambda_2$ are considered.

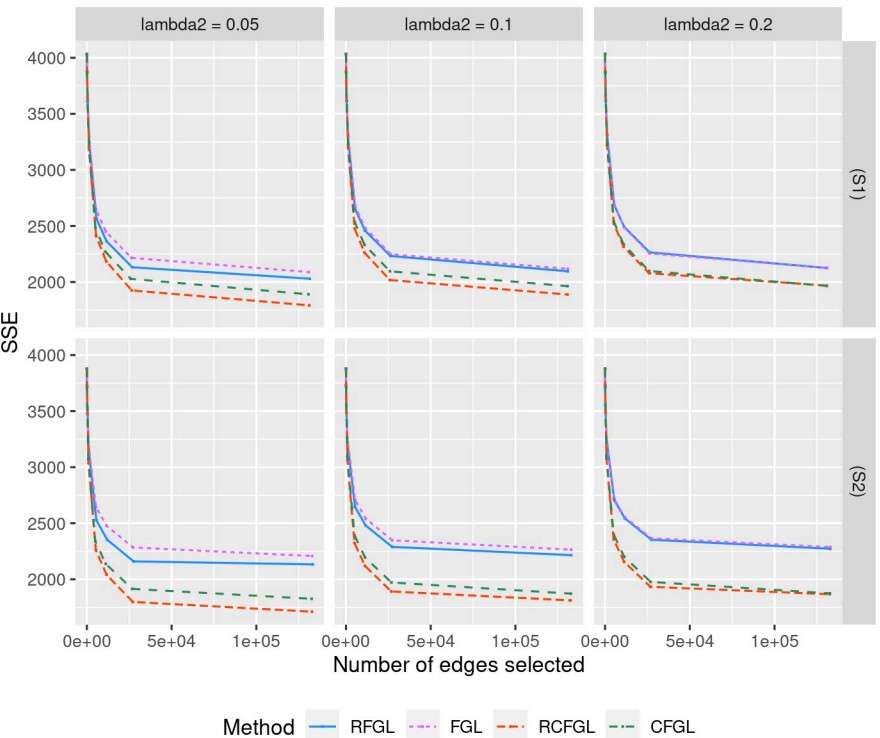

**Fig 3. Comparison of edge-weight estimation performance for simulations with three conditions.** Top and bottom rows respectively correspond to the SSE of methods in scenario (S1) and scenario (S2). The $x$ and $y$ axes respectively correspond to the total number of edges detected and SSE for different values of $\lambda_1$. Three different values of $\lambda_2$ are considered.

edge detection and the SSE of the methods. Fig 4 shows the comparison of the run-times of different methods. The comparison is demonstrated across low to high values of $\lambda_1$ since it controls how dense the networks will be and a denser network may take more time to be estimated.

Since CFGL is a condition adaptive extension of FGL and RCFGL is a condition adaptive extension of RFGL, it would be sensible to compare the methods pairwise. In scenario (S1), all the methods had nearly identical precision-recall curves for edge detection (Fig 2), especially for smaller values of $\lambda_2$. However, in scenario (S2) where the assumption of same level of similarity across all the pairs of conditions was violated, CFGL and RCFGL respectively achieved better precision-recall curves than their non-condition adaptive counterparts FGL and RFGL. In addition CFGL and RCFGL showed significantly lower SSE compared to FGL and RFGL in both the scenarios for all three values of $\lambda_2$ (Fig 3). This illustrates the advantage of the condition adaptive methods over the simpler ones especially when some pairs of conditions share different levels of similarity. CFGL had significantly higher run-time compared to all the other methods, whereas RFGL and RCFGL took just fractions of that time (Fig 4). RFGL was notably faster than FGL. So, when there are many genes and a large number of conditions, RFGL can be used over FGL for a much faster network exploration. RCFGL was also faster than FGL, such that one can perform a condition adaptive network estimation in a similar amount of time taken by a non-condition adaptive network estimation model such as FGL.

**Simulation with four conditions.** Next, we evaluated the performance of RCFGL in scenarios with four conditions. In this case the CFGL $R$ package was not usable and was omitted

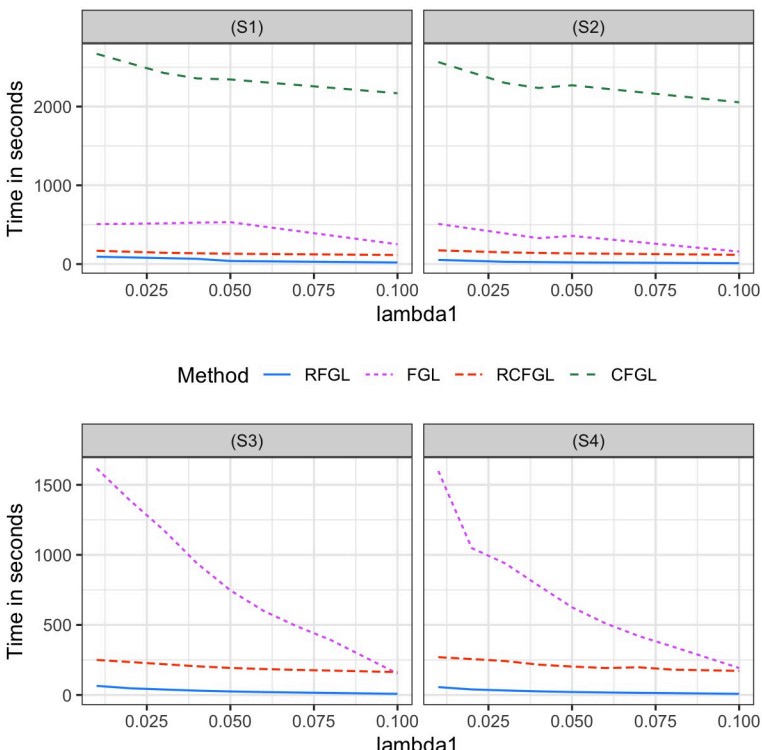

**Fig 4. Comparison of run-time for simulations with three and four conditions.** Top row corresponds to the run-times in seconds of different methods in scenario (S1) and scenario (S2). Bottom row corresponds to the run-times in scenario (S3) and scenario (S4). For the *x*-axis $\lambda_1$ is varied from low to high generating increasingly sparser networks. For each value of $\lambda_1$, the average run-time over three values of $\lambda_2$ is reported.

from comparison. Scenario (S3) is close to the assumption of FGL and RFGL that all the networks share same level of similarity, whereas (S4) violates that assumption. RCFGL had consistently better precision-recall curves compared to the other methods in all the scenarios (Fig 5). For larger values of $\lambda_2$, the precision-recall curve of RFGL was very close to the curve of FGL. RCFGL also had significantly lower SSE compared to RFGL and FGL in both the scenarios for all the values of $\lambda_2$ (Fig 6). RCFGL took significantly lower run-time compared to FGL, especially for smaller $\lambda_1$'s (Fig 4). It reaffirmed our earlier point that using RCFGL one can perform a condition adaptive network estimation even faster than a non-condition adaptive network estimation model such as FGL.

**Sensitivity with respect to ordering of the conditions.** Next, we studied the impact of different orderings of the conditions on RCFGL. Note that in (S7), two conditions 'far' from each other (conditions 1 and 4) had the same networks, whereas in (S6), two conditions relatively closer (conditions 1 and 3) had the same networks. We compared the performance of RCFGL with 'incorrect' ordering of conditions ((1, 2, 3) for (S5) and (1, 2, 3, 4) for (S6) and (S7)) with RCFGL with 'correct' ordering of conditions ((1, 3, 2) for (S5), (1, 3, 2, 4) for (S6), and (1, 4, 2, 3) for (S7)). In the plots, we referred to the latter as RCFGL-C and it was expected to perform the best. We compared regular RCFGL and RCFGL-C with FGL because it was the only method that would not be affected by the ordering (CFGL would also not be affected but could not be used with 4 conditions). Fig 7 displays the SSE for edge-weight estimation. In (S5), RCFGL and RCFGL-C had similar SSE values except for the smallest $\lambda_2$. In both (S6) and (S7), for smaller values of $\lambda_2$, RCFGL-C had noticeably better

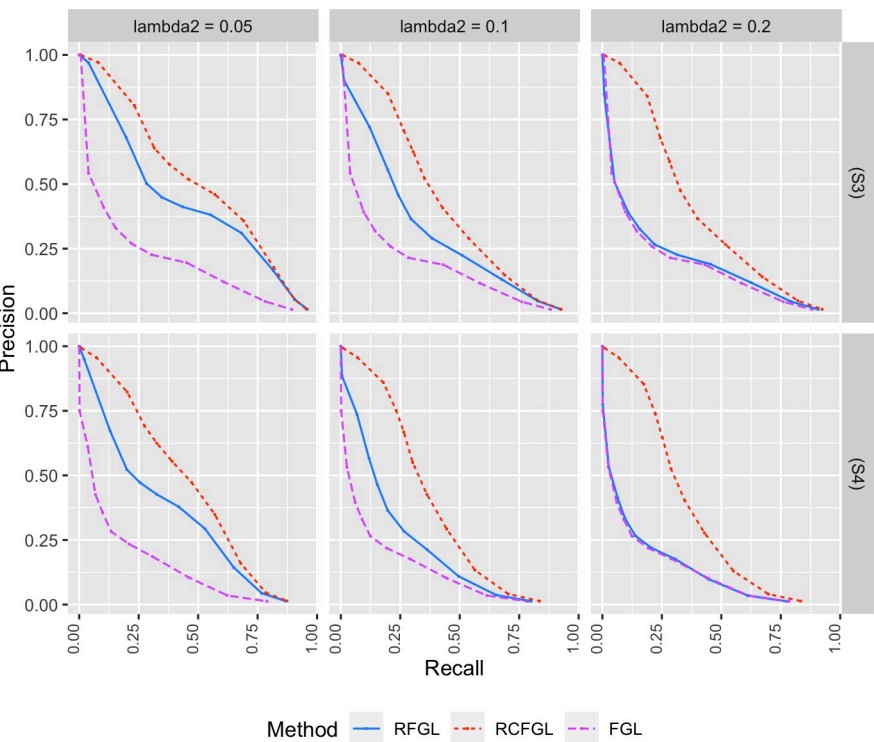

**Fig 5. Comparison of edge detection performance for simulations with four conditions.** Top and bottom rows respectively correspond to the ROC curves in scenario (S3) and scenario (S4). The *x* and *y* axes respectively correspond to false positive rate (FPR) and true positive rate (TPR) for different values of $\lambda_1$. Three different values of $\lambda_2$ are considered.

SSE compared to RCFGL. However, RCFGL had better SSE compared to FGL for all the values of $\lambda_2$. Therefore, it could be concluded that the effect of 'incorrect' ordering will have more of an impact with more than three conditions, particularly when two conditions 'far' from each other are similar.

The procedure we discussed in Effect of ordering of the conditions section to identify the ordering was able to detect the correct order in every scenario i.e., it placed the similar conditions side by side, the conditions (1, 3) in (S5), (S6) and (1, 4) in (S7). Therefore, using the proposed order-detection procedure, we achieved the best possible performance of RCFGL-C.

## Real data analysis

In the real data, the true networks are unknown and thus, we focused on checking the consistency of the estimated networks by different methods and their run-times, followed by a brief gene set enrichment analysis. The medial prefrontal cortex regions regions IL and PL are anatomically closer, and have been found to be similar in terms of overall structure and regulatory functions in many studies [68–70]. In our dataset as well, IL and PL were found to be similar in terms of the gene-expression based on hierarchical clustering (S2 Fig) compared to LHB. The condition adaptive methods, such as CFGL and RCFGL, were expected to detect more edges common between IL and PL compared to FGL and RFGL.

**Time comparison.** FGL and CFGL both would be computationally infeasible to run on all of 15,421 genes. So, we focused on smaller sets of genes obtained by pruning based on coefficient of variation (CV) [71]. To prune, we concatenated the gene expression data from all the

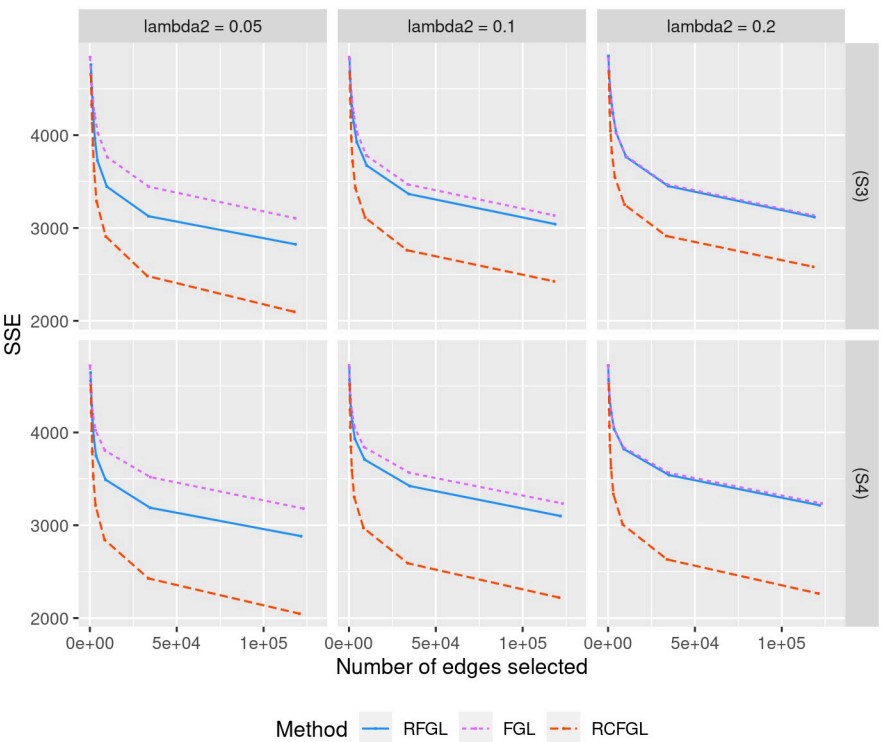

**Fig 6. Comparison of edge-weight estimation performance for simulations with three conditions.** Top and bottom rows respectively correspond to the SSE of methods in scenario (S3) and scenario (S4). The $x$ and $y$ axes respectively correspond to the total number of edges detected and SSE for different values of $\lambda_1$. Three different values of $\lambda_2$ are considered.

regions and computed the CV (ratio of mean to SD) of every gene. Next, we removed the genes which had CV less than a certain cut-off from the analysis. For example, removing the genes with CV < 0.02 left us with 1,106 genes, whereas removing the genes with CV < 0.04 left us with only 201 genes. We considered five such CV cut-offs, 0.015, 0.02, 0.025, 0.03 and 0.04. RFGL and RCFGL consistently took just fractions of the time taken by FGL and CFGL (Table 2). For the CV cut-off of 0.015, there were 4706 genes in the sample. In that case, we only reported the time taken by RFGL and RCFGL since both FGL and CFGL would be taking an exorbitant amount of time (more than 10 hours) to converge. It should also be mentioned that we had applied RFGL and RCFGL on the full dataset with 15,421 genes using a much more powerful Dell PowerEdge R740XD server with Intel Xeon Gold 6152 2.1G X (2) CPU having 44 cores, and they respectively took around 4 and 7 hours.

**Comparison of RFGL and RCFGL.** Next, we compared the networks estimated by RFGL and RCFGL. We considered the set of 557 genes, obtained by pruning the full set of genes based on the CV cut-off of 0.025. To address the variability in the estimated networks, we repeated the following procedure 50 times. Each time, we randomly selected 500 genes from the set of 557 genes and estimated the networks using RFGL and RCFGL for $\lambda_1 = 0.01$ and three values of $\lambda_2$. To investigate brain-region specificity of the edges detected by the two methods, we partitioned the identified edges into seven mutually exclusive categories: LHB region only, IL region only, PL region only, LHB-IL shared, IL-PL shared, PL-LHB shared, and common between all regions. Fig 8 displays the box-plot of the proportion of edges detected by the two methods. As mentioned earlier, we expected the regions IL and PL to

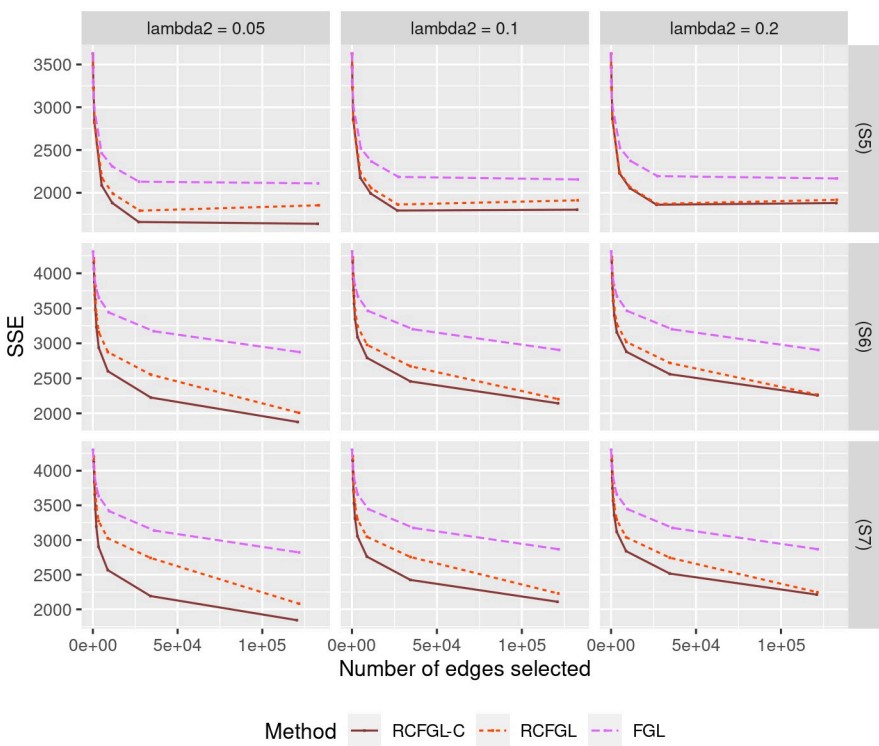

**Fig 7. Comparison of edge-weight estimation performance for simulations studying effect of ordering.** Top row corresponds to scenario (S5) which has three conditions, and the next two rows correspond to scenario (S6) and (S7) each of which has four conditions. The $x$ and $y$ axes respectively correspond to the total number of edges detected and SSE for different values of $\lambda_1$.

share more edges compared to region LHB and condition adaptive methods should be better at capturing that. Consistent with the expectation, we noticed that RCFGL detected more IL-PL specific edges compared to RFGL with the difference becoming increasingly apparent as $\lambda_2$ increased. We performed a pairwise t-test to determine statistical significance of this observation. For the three values of $\lambda_2$, 0.02, 0.03 and 0.04, the respective $p$-values were 0.01, 6e-12 and 2e-16, which indicated increasing statistical significance of the difference between the numbers of IL-PL specific edges detected by RFGL and RCFGL. RCFGL also detected more LHB specific edges. RFGL produced more edges common between all regions. A large value of $\lambda_2$ implies imposing a very high similarity penalty that would force the estimated networks of the three regions to be very close to each other. Thus, as we increased $\lambda_2$, both the methods

**Table 2. The run-times of different methods (in seconds) with the genes left after pruning based on different CV cut-offs.** $\lambda_1$ and $\lambda_2$ were respectively kept at 0.01 and 0.02. The mark "X" means that we could not run those methods due to inordinate amount of time required.

| CV cut-off (# genes left) | RFGL | FGL | RCFGL | CFGL |
|---|---|---|---|---|
| 0.040 (201) | 2 | 20 | 14 | 36 |
| 0.030 (376) | 7 | 102 | 27 | 145 |
| 0.025 (557) | 15 | 335 | 64 | 470 |
| 0.020 (1106) | 62 | 1468 | 172 | 2548 |
| 0.015 (4674) | 1511 | X | 3237 | X |

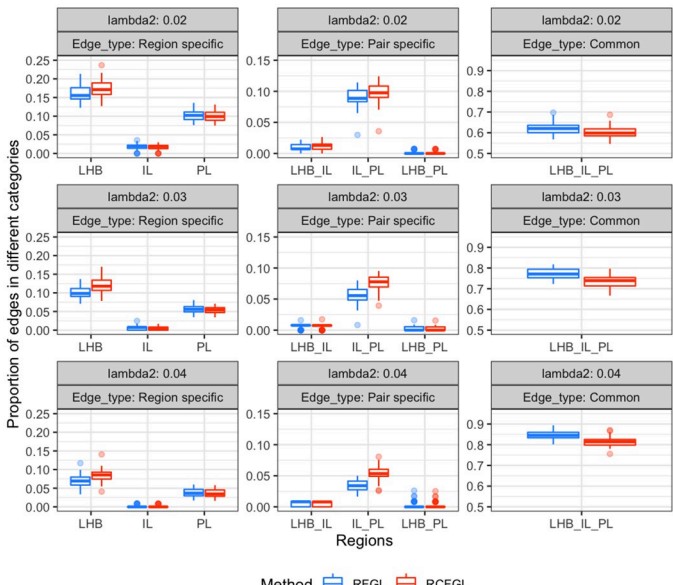

**Fig 8. Comparison of edge detection by RFGL and RCFGL in real data.** The *y* axis corresponds to the proportion of edges in seven mutually exclusive categories out of all the edges. The categories are coupled and displayed in three columns. The first column has edges specific to different regions. The second column has edges specific to different pairs of regions, and the third has edges common to all the regions. The rows from top to bottom respectively correspond to three different values of $\lambda_2$, 0.02, 0.03 and 0.04.

produced more edges common between all three regions and fewer edges specific to a single region or a pair of regions.

**Comparison of RCFGL and CFGL.** Next, we compared the performance of RCFGL to CFGL on the set of 557 genes (CV cutoff < 0.025). We kept $\lambda_1$ at 0.01 and varied $\lambda_2$ from low to high. We compared the top *Z* edges detected by the two methods to investigate the degree of agreement. We used the measure prop(*Z*) from Eq 5 for several values of *Z*. The top edges detected by RCFGL and CFGL matched by a great degree (prop(*Z*) > 0.85) in all the cases (Table 3). The agreement expectedly increased as $\lambda_2$ increased because for large values of $\lambda_2$, the difference between the penalty terms of RCFGL and CFGL becomes minimal, making them theoretically very close.

**Gene set enrichment analysis.** Our next goal was to identify biological functions of the hub-genes of the estimated networks by RCFGL using the methodology described in section Measures for evaluating performance. We ran RCFGL on the set of 1106 genes (CV cutoff < 0.02) with $\lambda_1 = 0.01$ and varying values of $\lambda_2 = 0.001, 0.0025, 0.005, 0.01$ and 0.05. Refer to S1 File for the full list of genes. The lowest value of AIC was observed for $\lambda_2 = 0.01$ and

**Table 3. Proportion of overlap of the top edges detected by RCFGL and CFGL.** The cells correspond to prop(*Z*) for varying values of *Z* and $\lambda_2$.

| *Z* | $\lambda_2 = 0.01$ | $\lambda_2 = 0.015$ | $\lambda_2 = 0.02$ |
|---|---|---|---|
| 100 | 0.86 | 0.94 | 0.95 |
| 300 | 0.89 | 0.93 | 0.96 |
| 500 | 0.92 | 0.95 | 0.97 |
| 700 | 0.94 | 0.94 | 0.97 |

**Table 4. Top pathways detected by the enrichment analysis of the hub-genes whose degree decreased from IL and PL to LHB.**

| Enrichment FDR | # Hub-genes in Pathway | # Background Genes in Pathway | Fold Enrichment | Pathway |
|---|---|---|---|---|
| 0.005 | 3 | 18 | 48.8 | Response to corticosterone |
| 0.009 | 3 | 27 | 34.87 | Response to mineralocorticoid |
| 0.009 | 3 | 108 | 30.51 | Response to calcium ion |
| 0.035 | 3 | 113 | 18.78 | Response to glucocorticoid |
| 0.048 | 3 | 165 | 13.56 | Response to ketone |
| 0.048 | 3 | 127 | 15.25 | Response to corticosteroid |
| 0.048 | 2 | 58 | 40.68 | Cellular response to calcium ion |
| 0.048 | 3 | 189 | 13.56 | Response to alcohol |

we interpreted the corresponding network estimates. There were 11 genes that were highly connected in the medial prefrontal cortex regions IL and PL but lost that connectivity in the LHB. These eleven genes were highly enriched (FDR < 0.01) for "Response to corticosteroid" (GO: 0031960) and similar GO terms. This follows what is known about the differences between these two brain regions. The medial prefrontal cortex has a well-established role as one of the primary sites for stress regulation and as a key site for glucocorticoid actions [72]. In contrast, the LHB is further downstream and receives stress-related signals from the medial prefrontal cortex [73]. Likewise, 57 genes were highly connected in the LHB but lost that connectivity in the medial prefrontal cortex. These genes were enriched for "Intestinal immune network for IgA production" (KEGG Pathway). Microglial cells are the dominant immune-related cell type in brain. Several studies recently have demonstrated the heterogeneity of these cells across brain regions [74, 75], so it is not surprising that the connectivity of genes related to immune response differed across brain regions. Additional enrichment results can be found in Tables 4 and 5, respectively listing the top pathways detected using the two sets of hub-genes: the genes whose degree decreased from IL and PL to LHB and the genes whose degree increased. Refer to S2 and S3 Files for the names of these two sets of genes. Fig 9 shows the estimated networks between these two sets of genes in the three brain regions. The networks corresponding to the regions IL and PL looked more similar to each other than LHB.

## Discussion

We present a method, named rapid condition adaptive fused graphical lasso (RCFGL) for estimating gene co-expression networks of multiple conditions jointly. Similar to an existing

**Table 5. Top pathways detected by the enrichment analysis of the hub-genes whose degree increased from IL and PL to LHB.**

| Enrichment FDR | # Hub-genes in Pathway | # Background Genes in Pathway | Fold Enrichment | Pathway |
|---|---|---|---|---|
| 0.026 | 4 | 68 | 48.8 | Leishmaniasis |
| 0.026 | 4 | 108 | 34.87 | Toxoplasmosis |
| 0.026 | 4 | 91 | 30.51 | Staphylococcus aureus infection |
| 0.026 | 5 | 167 | 18.78 | Tuberculosis |
| 0.026 | 4 | 91 | 13.56 | Systemic lupus erythematosus |
| 0.026 | 5 | 72 | 15.25 | Antigen processing and presentation |
| 0.026 | 3 | 43 | 40.68 | Intestinal immune network for IgA production |
| 0.026 | 3 | 26 | 13.56 | Asthma |
| 0.033 | 3 | 60 | 13.56 | Inflammatory bowel disease |
| 0.033 | 5 | 70 | 13.56 | Viral myocarditis |

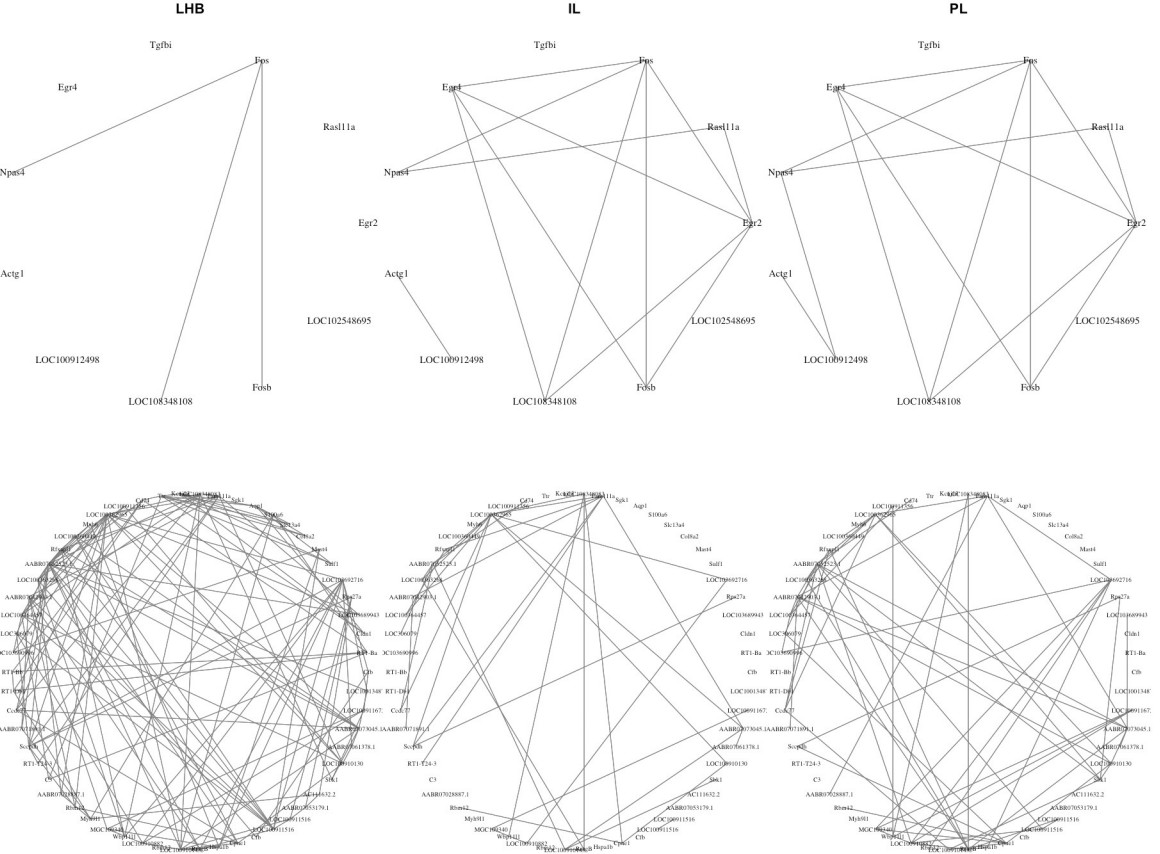

**Fig 9. The networks between the hub-genes whose degree changed from IL and PL to LHB.** The top row corresponds to the genes whose degree decreased from IL and PL to LHB, while the bottom row corresponds to the genes whose degree increased.

method named condition adaptive fused graphical lasso (CFGL), we compute data-driven weight terms between every pair of conditions storing information about pair-specific co-expression patterns. We include the weight terms in a sequential fused lasso penalty, a penalty earlier considered in a method named fused multiple graphical lasso (FMGL). As CFGL is interpreted as a condition adaptive extension of the method fused graphical lasso (FGL), RCFGL can be interpreted as a condition adaptive extension of FMGL. Unlike CFGL, RCFGL is computationally much faster and can be used to analyze more than three conditions together. As we have seen in the simulation studies and real data analysis, the performance of RCFGL and CFGL are very comparable. Both the methods outperform non-condition adaptive methods FMGL (referred to as RFGL in the figures) and FGL. We have demonstrated how fast RCFGL is compared to CFGL and even FGL in most of the cases.

We considered simulation scenarios with both three and four conditions. With three conditions, RCFGL and CFGL both achieved better precision-recall curves and smaller sum of squared error (SSE) than the non-condition adaptive methods FMGL and FGL, especially when there was a different level of similarity between the conditions. Furthermore, RCFGL took just a fraction of time taken by CFGL. With four conditions as well, RCFGL achieved superior performance than both FMGL (RFGL) and FGL, in addition to being computationally much faster than FGL. As an example real data analysis, we analyzed gene expression data from three brain regions, two medial prefrontal cortex regions IL and PL and another region

named LHB, from a heterogeneous stock panel of rats. We first compared the time taken by different methods to estimate the co-expression networks with varying sets of genes, showing again the computational feasibility of RCFGL. Then, we compared the performance of FMGL (RFGL) and RCFGL. The results demonstrated that network estimation was likely superior in the latter since it could detect more edges shared only between IL and PL, two medial prefrontal cortex regions that are expected to be more similar compared to the other region LHB. Finally, we performed enrichment analysis with the hub-genes of the estimated networks by RCFGL whose degree decreased from IL and PL to the LHB region, finding association with stress regulation and glucocorticoid actions.

Even though our method is developed for the purpose of estimating gene co-expression networks, it can be applied to any dataset that requires joint estimation of multiple networks and would benefit from taking into account condition specificity. In this paper, we have considered a maximum of four conditions. But, the run-time of RCFGL is approximately linear with respect to the number of conditions which makes it scalable for any number of conditions as long as the results remain interpretable. However, one limitation of both RCFGL and CFGL is that the weight-terms which capture information about pair-specific co-expression patterns, are binary. That is the weight for an edge between a pair of conditions takes value 1 if the edge is expected to be present in both the conditions and 0 otherwise. Future extensions will allow for continuous valued weight terms that will allow for more flexibility and can potentially improve performance.

RCFGL is implemented in the form of an open-source software package based on *C* and *Python*, available with a detailed *Jupyter* notebook at this link, https://github.com/sealx017/RCFGL. The package also implements the non-condition adaptive method, FMGL (RFGL). Note that the authors of FMGL provide a package that requires *MATLAB* and thus, it is not entirely open-source. Our package can be used as an open-source alternative of their package. The package also includes several tools for downstream analyses such as comparing networks across conditions and visualizing common or pair-specific networks. The code used to generate and analyze the datasets of the simulation studies are also provided with detailed documentation.

## Disclosure

The content is solely the responsibility of the authors and does not necessarily represent the official views of the National Institutes of Health.

## Supporting information

**S1 Fig. Workflow of the proposed method.** Expression data of multiple ($p$) genes are available in multiple ($K$) conditions at the start. In the next step, pair-specific patterns of similarity and dissimilarity between consecutive pairs of conditions are explored. In the final step the full model is fitted to jointly estimate all the networks using the proposed model.
(TIFF)

**S2 Fig. Hierarchical clustering based on the gene-expression data of three brain regions.** We concatenated the expression data of the 1106 genes (left after pruning based coefficient of variation (CV) cut-off of 0.02) and computed the Euclidean distance between each pair of brain regions. Next, hierarchical clustering was performed on the distance matrix revealing the order of similarity.
(TIFF)

**S1 Text. Proof of the theorem, connection between the penalty terms and quality control steps.** We provide the proof of the theorem for detecting block-diagonal structure in the precision matrices and derive the connections between the penalty terms used in different methods. We also list the quality control steps used in the real data pruning.
(PDF)

**S1 File. List of all the genes used in the enrichment analysis.** We provide the list of 1106 genes used in the enrichment analysis.
(CSV)

**S2 File. List of the hub-genes whose degree decreased from IL and PL to LHB.** We provide the list of the genes whose degree were lower in the networks of IL and PL than the network of LHB.
(CSV)

**S3 File. List of the hub-genes whose degree increased from IL and PL to LHB.** We provide the list of the genes whose degree were higher in the networks of IL and PL than the network of LHB.
(CSV)

## Author Contributions

**Conceptualization:** Souvik Seal, Katerina Kechris.

**Formal analysis:** Souvik Seal.

**Funding acquisition:** Qunhua Li, Laura M. Saba, Katerina Kechris.

**Investigation:** Souvik Seal, Katerina Kechris.

**Methodology:** Souvik Seal, Qunhua Li, Elle Butler Basner, Laura M. Saba, Katerina Kechris.

**Project administration:** Katerina Kechris.

**Resources:** Laura M. Saba, Katerina Kechris.

**Software:** Souvik Seal.

**Supervision:** Katerina Kechris.

**Validation:** Souvik Seal, Elle Butler Basner.

**Visualization:** Souvik Seal, Katerina Kechris.

**Writing – original draft:** Souvik Seal, Katerina Kechris.

**Writing – review & editing:** Souvik Seal, Qunhua Li, Elle Butler Basner, Laura M. Saba, Katerina Kechris.

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
