## [Decision Letter · Decision Letter 0]

29 Jul 2022

Dear Dr. Seal,

Thank you very much for submitting your manuscript "RCFGL: Rapid Condition adaptive Fused Graphical Lasso and application to modeling brain region co-expression networks" for consideration at PLOS Computational Biology.

As with all papers reviewed by the journal, your manuscript was reviewed by members of the editorial board and by several independent reviewers. In light of the reviews (below this email), we would like to invite the resubmission of a significantly-revised version that takes into account the reviewers' comments.

We cannot make any decision about publication until we have seen the revised manuscript and your response to the reviewers' comments. Your revised manuscript is also likely to be sent to reviewers for further evaluation.

Sincerely,

Inna Lavrik

Associate Editor

PLOS Computational Biology

Ilya Ioshikhes

Deputy Editor

PLOS Computational Biology

Reviewer's Responses to Questions

**Comments to the Authors:**

Reviewer #1: The review is uploaded as an attachment

Reviewer #2: This manuscript presents an algorithm to estimate gene co-expression networks from multiple experimental conditions. It builds up on top of prevoius work from the same authors denoted by CFGL, based on graphical lasso regularization techniques, where they introduced an algorithm for the same purpose, but which was limited to three conditions. The presented approach in this manuscript denoted by RCFGL, tweaks CFGL, concretely the penalty term of the graphical lasso regularitzation technique, to enable the use of the algorithm on more than three conditions, and running faster than the previous one. From that perspective, the contribution is somewhat incremental, in the sense that it does not introduce any substantially novel idea or technique.

The main technical contribution is using a sequential fused lasso penalty as in fused multiple graphical lasso (FMGL), instead of a pairwise one, which allows one to use more efficient and precise existing solvers for the maximizing the penalized log-likelihood.

The simulations provided show that RCFGL improve the TPR and FPR, as well as it running performance, over CFGL, while the analysis with real expression data from three brain regions, demonstrate that RCFGL finds more common co-expression relationships between the two more similar brain regions, than

FMGL.

Specific comments

1. In the financial disclosure, "Q.L. was National Institute of General Medical Sciences.. " should be "Q.L. was supported by ..". In the same section, the statement about the role of funders is probably wrong, because as it's written now, it seems that all funders play a role in the research, but from the descriptions of those roles, it seems that they correspond to the role of the authors. Likewise, the statement in the competing interests seems also wrong, the authors should have probably written that they declare that no competing interests exist.

2. In the data and code availability statement the authors declare that simulated data is available upon request. That's simply unacceptable from the perspective of this reviewer.

3. pg. 3 In "We use iterative ADMM algorithm .." an article is missing.

4. pg. 8 A theorem, Theorem 1, is given without proof.

5. pg. 8 In "For example, suppose there three conditions .." the verb "are" is

missing.

6. pg. 9 The specification of the ordering in which RCFGL is applied to each condition is made on the basis of minimizing the distance between every pair of sample covariance matrices derived from each corresponding condition. This distance is calculated here using the Euclidean distance, which is sensitive to large squared terms, which in this context may arise from the instability of these matrices derived from data with many more genes than available samples. This problem has been widely studied, see, for instance, Bryan et al. The comparison of sample covariance matrices using likelihood ratio tests, Biometrika, 1987, and more recent accounts in the field of quantitative genetics, such as in Steppan et al. Comparative quantitative genetics: evolution of the G matrix, Trends in Ecology & Evolution, 2002.

7. pg 11 The performance is evaluated in terms of TPR, FPR and ROC curves, where true positives (TPs) refer to predicted edges present in the real network and false positives (FPs) refer to predicted edges absent from that network. The decision to cast a TP or FP seems to be made on the basis of the comparison of the cells of the estimated precision matrix with the zero value. However, it is unclear whether this is decided using some tolerance or expecting an exact zero. The authors should clarify. Moreover, in the context where the number of true negatives (TNs), i.e., the number of missing edges, is overwhelmingly larger than the number of TPs the FPR is not a good measure because FPR=FP/(FP+TN) and the very large number of TNs is going make FPR always small. A precision-recall curve, based on the rate of TPs over the total of predictions, i.e., PPV=TP/(TP+FP), is more suitable for such an unbalanced classification problem. In the same page, equation (5) is called the Jaccard Index (see https://en.wikipedia.org/wiki/Jaccard_index).

8. pg 16 The analysis of real transcriptomics data from brain is somewhat deceptive as there is no rationale behind finding very unspecific pathways among hub genes that are common to the conditions. The insight into the brain-specific molecular processes is very poor.

7. The software is provided at the GitHub repo https://github.com/sealx017/RCFGL with very poor installation instructions and little guidance on how to use it on a real dataset.

Reviewer #3: This work focuses on the important problem of coexpression network inference. Typically, coexpression networks are built using Spearman or Pearson correlation of gene expression data to calculate the edge weight between each gene pair. Although these networks have proven useful in many applications, Spearman/Pearson correlation methods introduce many indirect relationships between genes that may be “on or off” at the same time, but do not have a true functional relationship. Here, the authors have focused on a type of method that aims to reduce these spurious correlations. This results in a sparser network that contains more direct relationships. The authors have made three major contributions to an existing method of inferring sparse coexpression networks called condition adaptive fused graphical lasso (CFGL) that explicitly takes advantage of scenarios where multiple datasets are available, each capturing the transcriptome in a group of samples corresponding to a particular condition, and these conditions (and therefore the datasets) are meaningfully related to each other. First, they have introduced a sequential fused lasso penalty that can be applied when there is a natural sequence of biological conditions, which motivates keeping networks for adjacent conditions as similar as possible, second, this new penalty was combined with a previous idea of overlaying differential coexpression on an appropriate penalty term, and third, they released an efficient implementation of the method, allowing more than 3 conditions and a larger number of genes to be used. The proposed method is an excellent way to leverage biological relationships between related sample groups and contributes a useful implementation for the community, but there are some concerns regarding the evaluation scheme.

Major comments

1. Performance is evaluated several times using the ROC curve.

1.1. It would be great to see a precision-recall curve instead of a ROC curve. In gene coexpression inference, the number of positive examples are far fewer than the number of negative examples (i.e. the number of gene pairs with true edges between them are far fewer than the number of gene pairs without true edges between them), and precision-recall curves are a better measure of performance than ROC curves in these cases. Furthermore, in biological applications we are generally more interested in the positive edges. Cutoffs are often used in large networks, so specifically the top/most weighted/highest confidence edges are of larger interest. The precision-recall curve is sensitive to changing performance in calling positives correctly without adding a large amount of false positives.

1.2. Edges that are the wrong sign are still counted as true positives in this evaluation. It is unclear whether this is acceptable. There is value in value in being able to identify an edge/relationship that exists, but would struggle to call one method better than another if the method is consistently identifying relationships in the opposite manner in which they actually occur. It would be helpful to show how much the evaluation would change if edges of the wrong sign were not counted as true positives.

2. The comparisons across networks are interesting, but certainly expected due to the sequential penalty forcing IL and PL to be more similar to each other than to LHB. Currently, there is no direct evaluation of whether the differing edges between the networks reflect meaningful biological function. The gene enrichment analysis in the manuscript now does not address network-specific genes/edges. Hence, it would be best to move it from the Real data analysis results section to the Supplement and replace it with a new enrichment analysis that focuses on hub genes that were more prevalent in the lateral habenula (LHB) core than the infralimbic (IL) cortex and prelimbic (PL) cortex. As one expects LHB to differ more in function, it would be important to see if the genes that were more “hubby” (higher degree) in LHB compared to IL and PL are enriched for specific functions important for LHB (and vice versa for IL/PL compared to LHB). This could be done by subtracting the average degree of each gene in the IL/PL networks from its degree in the LHB networks and doing enrichment on the genes that meet some threshold for LHB - IL/PL degree.

3. Coexpression networks are often built using pairwise Pearson/Spearman correlations. These kinds of networks tend to be a measure of which genes are “on and off” together, but do not necessarily imply a causal or even functional relationship between them. The proposed method instead calculates a precision matrix, which generally should not contain edges between genes that are conditionally independent. This means, as an example, that one should not expect two genes that are turned on by the same transcription factor to have an edge between them.

3.1. It would be good to include Pearson/Spearman-based coexpression networks as baselines in all the analyses.

3.2. A bit more commentary on the difference between using this method and the commonly-used Pearson/Spearman correlation methods (or situations where it would be more/less advantageous) would be nice to orient people with less math background. This could be added to the introduction where “the co-expression network is constructed based on the dependence structure” is discussed (first paragraph).

Minor comments

1. There is no code to reproduce the simulated data. Releasing this code improves reproducibility and facilitates later comparisons. Alternatively, these simulated datasets could be added to the Github repository in Data.

2. In methods for the Real data (starting page 10/23):

2.1. The number of rats remaining “after extensive quality control” is given. What quality control measures are used here? The sentence being referred to is this: “After extensive 285 quality control, 83 rats remained with RNA-Seq from the lateral habenula (LHB) core, 286 84 rats with data from the infralimbic (IL) cortex, and 82 rats with data from the 287 prelimbic (PL) cortex [54,55].”

2.2. Only 64 rats were used in the final analysis, a supplemental table with sample/run identifiers would be helpful for reproducibility.

**Have the authors made all data and (if applicable) computational code underlying the findings in their manuscript fully available?**

Reviewer #1: **No: **Unfortunately, the authors did not attach any code to analyze the real data. A precise criterion for the selection of the original 15421 genes is not clear. There is also no list of the 557 genes used for gene enrichment analysis. The gene enrichment analysis itself was carried out using an online tool and is not sufficiently documented. I gave respective recommendations in the text of the review.

Reviewer #2: **No: **The authors say in their Data and Code Availability that "Simulated data is available upon request".

Reviewer #3: None

PLOS authors have the option to publish the peer review history of their article (what does this mean?). If published, this will include your full peer review and any attached files.

Reviewer #1: No

Reviewer #2: No

Reviewer #3: No
---

## [Decision Letter · Decision Letter 1]

24 Nov 2022

Dear Dr. Seal,

We are pleased to inform you that your manuscript 'RCFGL: Rapid Condition adaptive Fused Graphical Lasso and application to modeling brain region co-expression networks' has been provisionally accepted for publication in PLOS Computational Biology.

Best regards,

Inna Lavrik

Academic Editor

PLOS Computational Biology

Ilya Ioshikhes

Section Editor

PLOS Computational Biology

Reviewer's Responses to Questions

**Comments to the Authors:**

Reviewer #1: I enjoyed reading the new revision of this article. I am completely satisfied with the way the authors changed the text and took into account all my questions and comments.

Minor comments:

1) The authors refer to dataset GSE173141 as source data. This superseries consists of the GSE173136-GSE173140 series. The raw data can be easily downloaded via this link through the SRA Run Selector. Nevertheless, the authors also attached processed (supplementary) data to each series. I would advise to indicate this at least in the text of the paper (otherwise when you quickly go to dataset GSE173141 it is impossible to find the normalized data promised in the paper).

2) The article made a very good impression on me, not only with the content, but also with the quality of the figures. Perhaps, on the background of this high quality, my eye was caught by a little sloppiness in the captions on the figures. The authors use the labels "lambda1, lambda2," whereas the text of the article uses more elegant Greek notations.

Reviewer #2: The authors have addressed all my concerns and I'm satisfied with the revised version of the manuscript. I only have a minor comment to make with respect to the author's response to one of the comments in my first review, concretely this one:

> pg. 3 In "We use iterative ADMM algorithm .." an article is missing.

> Reply: We thank the reviewer for the comment. We have now added the reference, “ Boyd et al. (2011) [2]”

I didn't mean to add a citation to a scientific article, which is also good, but that an article before an singular noun was missing. This is happening in page 3 and, incidentally, the same sentence is correctly written, including an indefinite article, in page 6, where it says "We use an iterative ADMM algorithm".

**Have the authors made all data and (if applicable) computational code underlying the findings in their manuscript fully available?**

Reviewer #1: Yes

Reviewer #2: Yes

PLOS authors have the option to publish the peer review history of their article (what does this mean?). If published, this will include your full peer review and any attached files.

Reviewer #1: No

Reviewer #2: No

---

## [Editor Report · Acceptance letter]

20 Dec 2022

PCOMPBIOL-D-22-00194R1 

RCFGL: Rapid Condition adaptive Fused Graphical Lasso and application to modeling brain region co-expression networks

Dear Dr Seal,

I am pleased to inform you that your manuscript has been formally accepted for publication in PLOS Computational Biology. Your manuscript is now with our production department and you will be notified of the publication date in due course.

With kind regards,

Zsofi Zombor
